# Microfracture- and Xeno-Matrix-Induced Chondrogenesis for Treatment of Focal Traumatic Cartilage Defects of the Knee: Age-Based Mid-Term Results

**DOI:** 10.3390/healthcare11222995

**Published:** 2023-11-20

**Authors:** Francesco Allegra, Aurelio Picchi, Marco Ratano, Stefano Gumina, Andrea Fidanza, Giandomenico Logroscino

**Affiliations:** 1Unit of Orthopaedics and Sport Medicine, ICOT, 04100 Latina, Italy; francescoallegra@tin.it; 2Unit of Orthopaedics, Department of Life, Health and Environmental Sciences, University of L’Aquila, 67100 L’Aquila, Italygiandomenico.logroscino@univaq.it (G.L.); 3Department of Anatomy, Histology, Legal Medicine and Orthopaedics, Polo Pontino, Sapienza University of Rome, 00161 Rome, Italy

**Keywords:** chondral defects, chondrogenesis transplantation, knee arthritis, patellofemoral joint, MOCART, AMIC

## Abstract

The aim of this study was to investigate clinical and instrumental outcomes of the autologous matrix-induced chondrogenesis (AMIC) technique for the treatment of isolated traumatic condyle and femoropatellar cartilage lesions. A total of 25 patients (12 males, 13 females, mean age 47.3 years) treated between 2018 and 2021 were retrospectively reviewed and subdivided into two groups based on age (Group A, age < 45 years; Group B, age > 45 years). A clinical evaluation was performed using the International Knee Documentation Committee (IKDC), Lysholm score and Visual Analogue Score (VAS). Cartilage regeneration was evaluated via magnetic resonance (1.5 Tesla) and classified according to a Magnetic resonance Observation of CArtilage Repair Tissue (MOCART) scoring system. At a minimum follow-up of 2 years, Group A patients obtained greater instrumental results in comparison to group B: in fact, the MOCART score was statistically significantly correlated with IKDC (r = 0.223) (*p* < 0.001) exclusively in group A. Nevertheless, a significant improvement in clinical functionality was shown in Group B (*p* < 0.001), demonstrating that this technique is safe, reproducible and capable of offering satisfactory clinical results regardless of age.

## 1. Introduction

Isolated articular cartilage lesions can lead to substantial patient morbidity and may progress to diffuse osteoarthritis [1]. The most common locations for chondral lesions in the knee are condyle and patellofemoral joint in athletes or active patients [2]. When a lesion occurs, the complex tissue structure and biomechanics of motion must be restored in order to solve the immediate problems and guarantee the long-term life of the joint [3,4].

There are many surgical options to treat symptomatic cartilage injury, depending on the extent of injury and the duration of symptoms. A wide variety of surgical procedures have been developed to address this problem [5]. Restoration techniques such as osteochondral autografts, mosaicplasty and osteochondral allografts attempt to replace the cartilage defect with host or donor cartilage in a single stage. Hangody et al. reported good to excellent clinical outcomes after osteochondral autografts in >90% of patients with defects measuring 1 to 5 cm^2^ at seventeen years of follow-up, although donor-site morbidity remains a concern [6].

Repair techniques or marrow stimulation techniques, such as abrasion arthroplasty, drilling and microfracture (MFx) penetrate the subchondral bone and induce the formation of fibrocartilage repair tissue [7]. Restoration with autologous chondrocyte implantation (ACI) attempts to generate hyaline or hyaline-like cartilage more effectively than MFx, but typically requires two surgical procedures [7]. Long-term follow-up (up to twenty years) after ACI has shown as much as a 92% rate of patient satisfaction, with sustained improvement in clinical outcomes and magnetic resonance imaging findings [8,9,10]. Although excellent short-term clinical outcomes have been demonstrated after marrow stimulation, the clinical durability of marrow-stimulated repair tissue has shown an objective and functional decline in long-term follow-up, especially in larger lesions [2,3,4,5]. Furthermore, third-generation ACI, matrix-assisted ACI (mACI), demonstrates a lower failure rate and greater improvement in patient-reported outcomes compared with MFx for focal chondral defects of the knee [11].

In 1998, Behrens et al. performed the first transplantation using a porcine collagen I/III matrix (Chondro-Gide, Geistlich Biomaterials, Wolhusen, Switzerland) in place of ACI. Instead of a periosteum flap, this was utilized as a substrate for the so-called matrix-associated autologous chondrocyte transplantation (MACT, MACI^R^, Verigen Transplantation Service, Copenhagen, Denmark). Later, Behrens validated his technique reporting thirty-eight patients with localized cartilage defects treated with MACT, resulting in significatively improved outcomes [12].

The Autologous Matrix-Induced Chondrogenesys (AMIC) technique combines microperforations with a xeno-matrix. This technique belongs to a MACT category and is a valid current surgical option with the aim to rebuild the cartilaginous tissue’s structural integrity [5,13,14]. In AMIC, the microfracturing is directly followed by the application of the biodegradable natural collagen type I/III membrane to host and hold the superclot generated by microfracturing [12]. Migliorini et al., in a systematic review, found better clinical outcomes from the AMIC procedure compared to mACI, despite additional research being required to validate this conclusion [14]. The age of patients undergoing these surgical procedures ranged from 14 to 70, according to the comprehensive systematic review of Steinwachs et al. [15].

In the current paper, we examined patients according to age (under or over 45 years) with the main objective of investigating clinical and instrumental outcomes in these two populations suffering from traumatic condylar and patellofemoral chondral lesions and treated with the AMIC technique. The first hypothesis of the authors is that better results can be obtained in younger patients. The second hypothesis is that the technique is safe and reliable even for patients over 45 years old.

## 2. Materials and Methods

In the sports medicine department, patients with all types of traumatic full-thickness (IV grade by Outerbridge classification) cartilage defects (hip, ankle, and knee) treated with AMIC between 2018 and 2021 were retrospectively reviewed.

The patients enrolled in this study met these inclusion criteria: isolated traumatic cartilage lesion of knee condyles and patellofemoral joint, IV degree according to Outerbridge classification [16], and area of the defect ranging from 1 cm^2^ to 5 cm^2^. The exclusion criteria were arthritis, age under eighteen and over sixty years, low-grade defects (I-II-III grade according to Outerbridge classification), cartilage defects of the tibial plateau, kissing lesions, previous surgery, axes defects, concomitant fractures, ligament or meniscal injuries, rheumatoid arthritis, body mass index (BMI) over 30, and other serious comorbidities that nevertheless represent a contraindication to elective surgical treatment. This study was conducted in accordance with the Declaration of Helsinki and approved by the Institutional Review Board. The patients enrolled expressed written informed consent to participate in the study and for the publications of their data for scientific purposes.

Patients who met the inclusion criteria were divided into two groups based on age, below or above 45 years old. The clinical outcomes were assessed using the International Knee Documentation Committee (IKDC) [17], Lysholm score [18], and VAS [19] scale systems at the pre-operatory time and at 12 and 24 months after surgery. Postoperatively, at twelve and twenty-four months of follow-up, magnetic resonance imaging (MRI) was performed to evaluate the quality of the repaired cartilage.

The MRI examinations were conducted with a protocol performed using a Magnetom Sola (Siemens Healthineers GmbH, Erlangen—Federal Republic of Germany) 1.5 Tesla permanent magnet, based on routine sequences (axial TSE T2-weighted, coronal T1-weighted, proton density fat-sat, sagittal thin-slice T2-weighted) and specific sequences for evaluating the chondral mantle. The thickness of this second category of sequences, which ranges from 0.6 mm to 1.2 mm, allows for reconstruction of the captured 3D images in various planes. Additionally, they allow for the evaluation of the trabecular force lines’ structural integrity and the identification of subchondral bone opening regions near the crater’s bottom. The Magnetic resonance Observation of CArtilage Repair Tissue (MOCART) scoring system was used [20].

The variables of the MOCART classification system are degree of defect repair and filling of the defect, integration to border zone, surface of the repair tissue, structure of the repair tissue, signal characteristics of the repair tissue, subchondral lamina, subchondral bone, adhesions attached to the repair side, and synovitis [20]. This score varies from zero to 100 and the higher the value, the greater the potential positive correlation with cartilage regeneration.

### 2.1. Surgical Procedure

The surgical procedure was conducted under tri-block, spinal or general anesthesia. It started with arthroscopy to assess the defect’s size, location and to identify any other intra-articular disorders. Subsequently, a minimally invasive arthrotomy, either lateral or medial, was performed to enhance visualization [21]. The damaged cartilage tissue was meticulously cleaned until it bled, and its edges were trimmed to the stable walls of the surrounding healthy cartilage. The subchondral bone was gently scraped using a curette. Microperforations were executed following Steadman’s technique [22]. The perforations were evenly distributed around the defective area, including its central site, ensuring a uniform pattern (Figure 1).

The required amount of residual bone bridges was between 3 mm and 5 mm apart, to avoid the risk of reduction of the bony biomechanical integrity and of its localized collapse [23]. The size of the defect was evaluated by using an aluminum sheet to determine its dimensions, fitting to the cartilage lesion by adaptation to the template, which matches its dimensions. The collagen membrane was then measured over the template, paying attention to cut it undersized to avoid dislocation after movement. Once the biodegradable membrane was implanted, the procedure was completed with peripheral fibrin glue fixation (Figure 2). We used a Chondro-Gide (Geistlich Biomaterials Italia srl, Thiene (VI) Italy) membrane. It is a bilayer membrane made of highly refined porcine collagen (type I/III), specifically developed for cartilage regeneration.

The minimally invasive surgical cut was closed in layers with standard techniques.

### 2.2. Rehabilitation Protocol

The post-operative management was standardized for all patients, allowing a maximum weight-bearing of 15–20 kg for 4–6 weeks. Thrombosis prophylaxis with low molecular weight heparin (LMWH) was administered universally.

For the initial 2 weeks following surgery, the maximum flexion of the knee was restricted to 30°. Assisted active physical therapies were employed, taking into consideration the weight and range of motion limitations.

Starting from the second week after surgery, patients began isometric quadriceps training, straight leg raises, and hamstring isometrics. Knee flexion was gradually increased to 60° and then 90° over the next 2 weeks. Subsequent rehabilitation programs incorporated progressive weight-bearing and a variety of mobilization exercises, along with electrotherapy for the leg muscles, proprioception exercises, and stair walking. Full weight-bearing was permitted in the sixth week after the surgical procedure.

### 2.3. Statistical Analysis

The International Business Machine corporation—Statistical Package for the Social Science software (IBM Corp. Released 2015. IBM SPSS Statistics for Windows, Version 23.0. Armonk, NY, USA: IBM Corp.) was used to examine the data.

Normality was checked with Shapiro–Wilk test. The statistical analysis was based on an estimated sample size of at least 24 subjects, which was calculated to be adequate to achieve 90% power to detect a large effect size (Cohen’s f: 0.40). Parametric tests were used for the analysis. A Chi-squared test was performed to evaluate the initial differences of the cartilage damage between the two groups. The correlation between clinical examination (IKDC) and instrumental examination (MOCART scoring system) was investigated using Pearson correlation coefficients. Independent-sample T-tests were used to check the differences between group A and group B. Additionally, repeated-measures analyses of variance (ANOVAs) were carried out, followed by paired samples T-tests, to evaluate the progression of pain reduction (VAS) and to assess progressive improvements of clinical and radiological scores. A *p* < 0.05 was considered significant.

## 3. Results

Our database consisted of 130 patients with several types of cartilage defects (Figure 3), of which 40 patients reported knee lesions (30.7%).

Finally, the study group consisted of 25 patients (12 male and 13 female with an average age of 47.3 years) with a median follow-up of 25.8 months (range 24–29 months).

The lesions were distributed as follows: twelve patellar lesions, five trochlear lesions, three lateral femoral condyle (LFC) lesions, and five on the medial femoral condyle (MFC).

The Group A patients (Table 1) consisted of twelve patients (5 male and 7 female, mean age 34 ± 7 years). The preoperative extent of the cartilage damage did not differ between the two groups (*p* = 0.03).

In Group A, the mean defect area measured intraoperatively was 2.7 ± 1.6 cm^2^. Five patellar defects, three troclear, one LFC and three MFC were treated. In this cohort, the IKDC preoperatively was 44 ± 4. After surgery, the score improved significantly (*p* < 0.001) at the first year 72 ± 2, and between the first and the second year 76 ± 2 (*p* = 0.001) (Figure 4a).

The Lysholm score preoperatively was 62 ± 3. After surgery, scores improved significantly at the first year 90 ± 4 (*p* < 0.001), and between the first and the second year 96 ± 3 (*p* < 0.001). Compared to the preoperative period, the pain measured via VAS score decreased at the first year, from 4 ± 1 to 1 ± 1 (*p* < 0.001), and at the second year was stable, at 1 ± 1. In group A at the first year, the MOCART score (Figure 4b) was 67 ± 4 and at the second year it was 73 ± 7 (*p* < 0.001). At the second year, the defect filling was complete in two patients, complete in more than 75% in five patients, between 50 and 75% in three patients, and between 25% and 50% in two patients; no patient experienced less than 25% defect filling.

The Group B patients consisted of 13 patients (8 male and 5 female, with an average age of 53 ± 3.5 years (Table 1). The mean defect area measured intraoperatively was 3.0 ± 1.2 cm^2^. Seven patellar defects, two troclear, two LFC and two MFC were treated. In this cohort the IKDC (Figure 4a) preoperatively was 45 ± 4; after surgery, IKDC improved significantly at the first year 69 ± 3 (*p* < 0.001), and between the first and the second year 72 ± 4 (*p* = 0.001). The Lysholm score preoperatively was 64 ± 2; after surgery, scores improved significantly at the first year 89 ± 3 (*p* < 0.001), and between the first and the second year 95 ± 4 (*p* = 0.04). Compared to the preoperative period, the pain measured via VAS score decreased at the first year, from 3 ± 1 to 2 ± 1 (*p* < 0.001) and remained stable at the second year, 1 ± 1. In group B at the first year, the MOCART score (Figure 4b) was 54 ± 4 and at the second year it was 56 ± 5 (*p* > 0.05). At second year, the defect filling was not totally complete in any patient, while it was filled for more than 75% in two patients, between 50 and 75% in six patients, between 25% and 50% in three patients, and less than 25% in two patients.

The preoperative and postoperative clinical and radiological scores of all patients are shown in Table 2.

We studied correlation between the clinical score (IKDC) and instrumental score (MOCART score): it was statistically significant within Group A, at the first (r = 0.223) (*p* < 0.001) and second year (r = 0.247) (*p* < 0.001).

Improvements in clinical assessment scores were statistically significant between the pre-operative period and the first year in the whole cohort, as well as between the first year and the second year. (IKDC *p* < 0.001; Lysholm *p* < 0.001; and VAS *p* < 0.001). Across the whole cohort, the MOCART score improved significantly between the first year, 60 ± 8, and the second year, 64 ± 10 (*p* < 0.001).

## 4. Discussion

To reduce the risk of loss of joint function and progression of osteoarthritis, surgical treatment is mandatory for patients with this type of injury [25]. Many studies have been conducted to evaluate and examine the results of patients undergoing surgery; however, patient populations were under 45 years old in the majority of papers, and for this reason we chose this cut-off to evaluate two different groups of patients based on age [26,27,28]. In our medium-term experience, Group A patients outperformed Group B in terms of instrumental outcomes and tissue regeneration. The patients in Group B had satisfactory clinical improvements, despite magnetic resonance imaging revealing that the cartilage tissue regeneration was of lesser quality.

Younger and more active patients, with a shorter duration of pre-operative symptoms, fewer surgical procedures prior to cartilage repair or restoration and no concomitant ligamentous instability, meniscal deficiency, or tibiofemoral or patellofemoral malalignment, can expect the best outcome regardless of technique [29]. The technique studied in middle-aged patients has been clinically advantageous; however, the regenerated cartilage tissue did not always prove to be optimal in our study. In four patients of group B (30%) at 1 year, the MRI showed a cystic resorption zone (Figure 5a) within the sublesional cancellous bone, with sclerotic borders and a signal consistent with granulomatous matrix. The same phenomenon with a centripetal healing margin was observed 2 years after surgery at the site of the previously described cystic resorption as a geoid appearance (Figure 5b).

This phenomenon was recorded in the MOCART and adversely affected the score outcome for these patients, due to granulomatous subchondral tissue, cystic alterations and the quality of the regenerated tissue in terms of hydration and tissue homogeneity.

Kusano et al. described that tissue filling in their cases (40 patients; mean age 35.6 years, range 23–43) was present but often not complete or homogenous when evaluated via MRI (MOCART) [30]. Schiavone-Panni et al. noted a reduction in the defect in 59% of cases (17 patients; mean age 39 years, range 22–52) and that it was not filled in all patient [31]. Pascarella et al. (19 patients; mean age 26 years, range 18–50) showed a significant reduction in the defect area, both in shape, filling, interface and subchondral oedema [32].

In our study, patients improved in clinical and instrumental scores at one and two years of follow-up, with more than 50% of defects filled in eighteen patients (72%). Statistical correlation was not demonstrated between clinical scores and imaging in the total sample. Gille [33] and Buda [34] observed that defect filling ranged from mild to complete and correlated with clinical outcomes. In a review and meta-analysis, Blackman [35] and De Windt [36] investigated the potential correlation between clinical and instrumental evaluation, but this was not demonstrated.

In our experience, in group A, there was a significant association between clinical and imaging outcomes. We found statistical correlation that supports age as a crucial factor (*p* < 0.001). However, our study did not show any statistical difference between the two groups in clinical outcomes, which on the contrary was supported in the study of Gille [37]. Older age was found by Filardo et al. to be a predictor of low clinical scores [38]; our data are not consistent with those: in fact, the 2 years follow-up clinical results were comparable for the two groups. The reason could be linked to the lack of long follow-up of our cohorts.

In regard to surgical approach, we opted for the mini-open technique, a method demonstrated to be comparable to arthroscopic AMIC [39]. While some studies suggest a faster recovery with arthroscopy, no significant differences have been observed within the first two years of follow-up [40]. Tan et al. reported similar outcomes between arthroscopic and mini-open approaches in a meta-analysis [41].

To secure the membrane, we utilized fibrin glue, a technique considered superior to suturing. Suturing with reabsorbable threads has been associated with cartilage damage resembling osteoarthritis [42]. Conversely, gluing has been proven effective and is a routinely practiced fixation method. This preference for gluing is supported by research showing enhanced chondrogenesis with biphasic carrier constructs comprising fibrin glue and the Chondro-Gide matrix [42]. Regarding graft selection, Binder et al. conducted a study comparing four different grafts, including Chondro-Gide, and found no significant differences [43]. Partially autologous fibrin glue is emerging as a preferred fixation method. This involves centrifuging a blood sample from the patient, mixing the resulting thrombin with allogenic fibrinogen. While utilizing bone marrow blood is another option, it poses potential challenges at the donor site [44].

This study suffers from some limitations. First of all, the limited number of patients enrolled, which, however, is in line with the other reports cited, given the uncommonness of the lesions treated with this technique. Another limitation of the study is the non-uniform anatomic distribution of injuries. Compared to an MFC lesion, a trochlear lesion receives a different load [45]. Windt et al. revealed some factors related to a better clinical outcome in patients treated using first-generation ACI or MACT or microfracture: defects located at the MFC and patients younger than 30 years, especially those with acute lesions [46].

## 5. Conclusions

The AMIC technique shows satisfying clinical results in all patients, regardless of age. According to MOCART score, the procedure looks to be more beneficial for patients under the age of 45 in term of regenerating tissue. However, a low MOCART value does not equate to a higher rate of failure or revision surgery. It is essential to conduct further randomized studies to evaluate whether the procedure is safe and reduces the risk of post-traumatic osteoarthritis in patients over 45.

## Figures and Tables

**Figure 1 healthcare-11-02995-f001:**
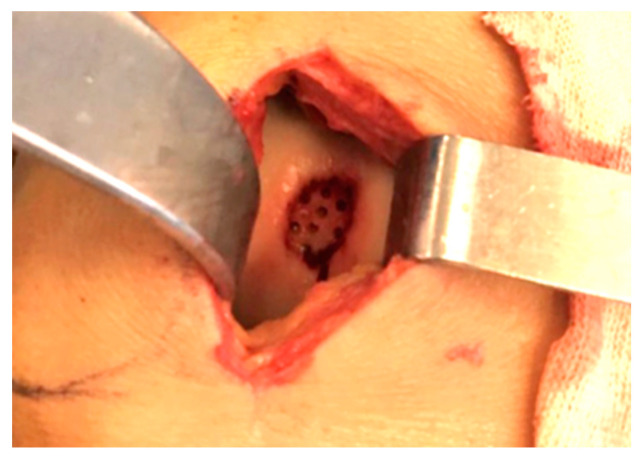
Steadman’s perforations in a defect of trochlea in a 24 y.o. male patient.

**Figure 2 healthcare-11-02995-f002:**
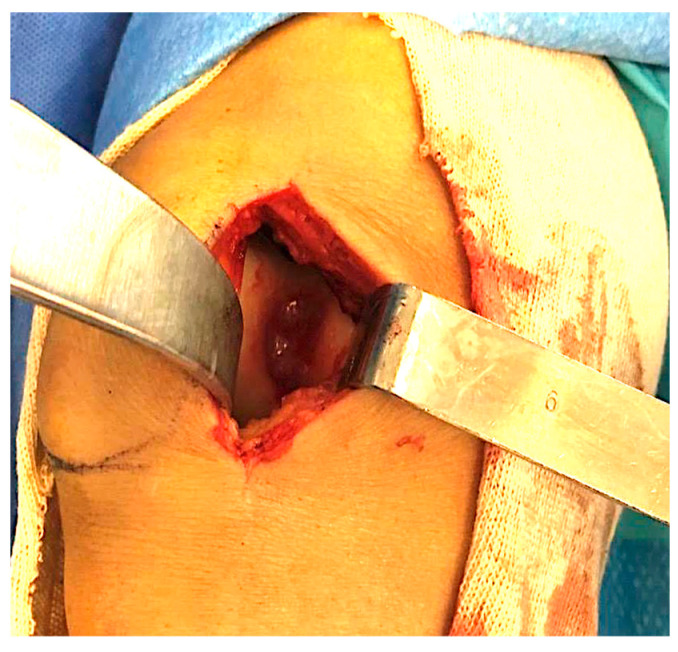
Final look of a defect of trochlea filled with biodegradable membrane fixed with fibrin glue.

**Figure 3 healthcare-11-02995-f003:**
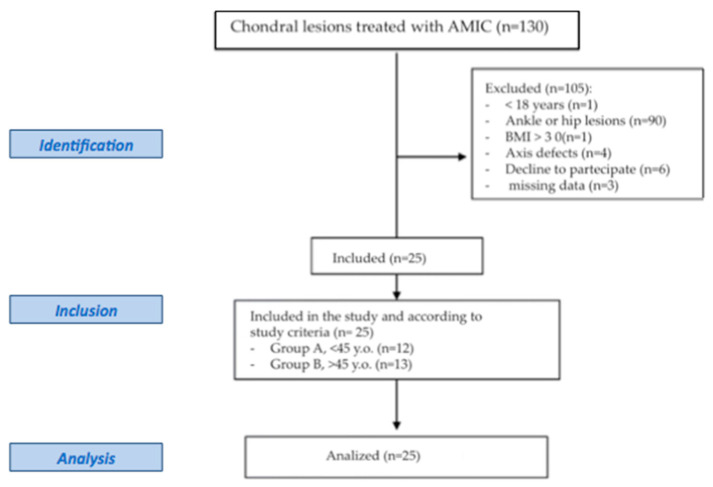
Flowchart of patients according to STROBE criteria [24].

**Figure 4 healthcare-11-02995-f004:**
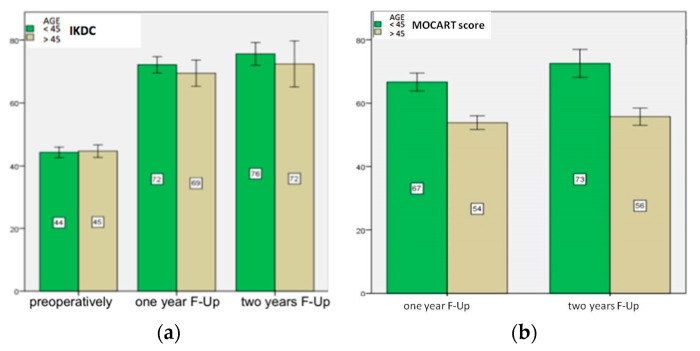
On “*y*-axis” scores of IKDC (**a**) and MOCART (**b**) are reported. At 1 and 2 years follow-up (F-Up), IKDC score showed satisfactory clinical results in both groups (*p* ≤ 0.001) (**a**); MOCART score at 1 and 2 years after surgery (**b**) showed better instrumental results in Group A (*p* < 0.001).

**Figure 5 healthcare-11-02995-f005:**
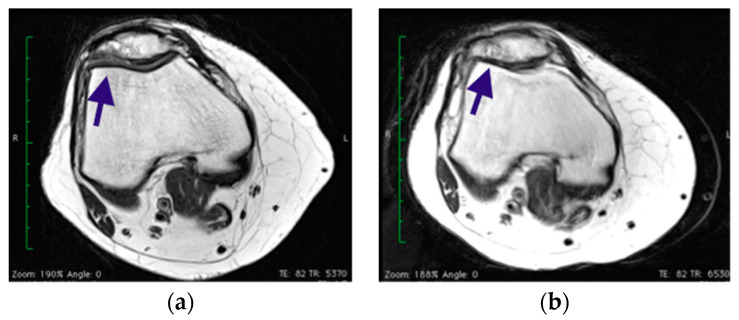
Patellar defect in over 45 patient shows cystic resorption zones after surgery (arrow). (**a**) MRI at 1 year of follow-up; (**b**) MRI at 2 years of follow-up.

**Table 1 healthcare-11-02995-t001:** Characteristics of the sample subdivided into two groups based on age. Mean age and mean area of defect are reported with standard deviation. M: male; F: female.

	n	Gender	Mean Age (y.o.)	Mean Area of Defect (cm^2^)
Group A<45 y.o.	12	5 M; 7 F	34 ± 7	2.7 ± 1.6
Group B>45 y.o.	13	8 M; 5 F	53 ± 3.5	3.0 ± 1.2

**Table 2 healthcare-11-02995-t002:** A comprehensive overview of clinical (IKDC, Lysholm and VAS) and radiological (MOCART) results of the two groups of patients; the reported *p*-values represent the trend from preoperative score to one-year follow-up, and from the first to the second year of follow-up (repeated-measures analysis of variance).

	Pre-Operative	One Year Follow-Up	Two Years Follow-Up
**IKDC** Group A *p*-value Group B *p*-value	44 ± 4	72 ± 2 <0.001	76 ± 2 =0.001
	45 ± 4	69 ± 3 <0.001	72 ± 4 =0.001
**Lysholm** Group A *p*-value Group B *p*-value	62 ± 3	90 ± 4 <0.001	96 ± 3 <0.001
	64 ± 2	89 ± 3 <0.001	95 ± 4 =0.04
**VAS** Group A *p*-value Group B *p*-value	4 ± 1	1 ± 1 <0.001	1 ± 1 -
	3 ± 1	2 ± 1 <0.001	1 ± 1 =0.2
**MOCART** Group A *p*-value Group B *p*-value	-	67 ± 4	73 ± 7 =0.02
	-	54 ± 4	56 ± 5 =0.6

IKDC: International Knee Documentation Committee; VAS: Visual Analogue Score; MOCART: Magnetic resonance Observation of CArtilage Repair Tissue scoring system. Values are reported as averages ± standard deviation.

## Data Availability

Data are available upon specific request to the corresponding author.

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
