# Peer review of "Microfracture- and Xeno-Matrix-Induced Chondrogenesis for Treatment of Focal Traumatic Cartilage Defects of the Knee: Age-Based Mid-Term Results"

_healthcare, 2023, doi:10.3390/healthcare11222995_

Round 1

Reviewer 1 Report

Comments and Suggestions for Authors

This manuscript describes an interesting case series of AMIC with mid-term follow-up.  The study’s inclusion and exclusion criteria were reasonable, and it is fortunate that the male/female breakdown is about the same in the older and younger cohorts. Very importantly, there was not a significant difference in average size of defects in the two groups.  The findings clearly show that the clinical outcome was essentially the same but that the imaging outcomes were different, with less defect filling and the occurrence of cystic resorption in the older age group (>45 yrs).  The conclusion that “Surgery looks to be more beneficial for patients under the age of 45” (line 300) is supported only by the lower MOCART score. The clinical outcome is the one that is important to a patient.  If AMIC itself does not increase the risk of developing PTOA above the risk presented by the lesion itself, then it seems to be beneficial for all patients in the 22-58 age range. The manuscript does not cite an inverse correlation between MOCART and risk of developing PTOA, which would influence interpretation of the results. The study could be more conclusive if groups A and B were compared to age-matched groups of similar patients who were treated conservatively.  That would allow for differentiation of the effects of the surgery from the natural progression or healing of the lesions.

Additional points to consider:

Line 84-85: The study was designed to investigate patellofemoral chondral lesions, but 8 out of 25 lesions were on the femoral condyles (3 lateral, 5 medial) – lines 175-177.  The manuscript should explain how these condylar lesions were distributed between groups A and B.

Lines 12-124: Were the lengths of the incisions any larger for patellar defects?  It seems that the articular side of the patella would be more difficult to access.

Lines 136-138: Be specific about the source of the collagen membranes.

Line 244: There should be some speculation about the cause of the cystic resorption zones in some group B patients. 

Figure 3 vertical axes should be labeled, even though they are described in the legend.

MRI scans in Figure 4 should be annotated to indicate precisely where the cystic resorption zones are located.

Comments on the Quality of English Language

Overall, the quality of the English is very good.

Reviewer 2 Report

Comments and Suggestions for Authors

References have to be updated to the most recent ones.

Abstract

Lines 37-38: please add also p-values.

Introduction

The introduction has to be improved also reporting data from more recent references.

Lines 51: References are lacking. Please add.

Line 55. Update this reference preferring a review of different surgical techniques.

Line 61: References are lacking. Please add.

Lines 69-77: are there more recent studies? Reporting only these studies (1994 and 1998), it seems that this technique is old. Please clarify.

Line 75: References are lacking. Please add.

Materials and methods

Line 89: please specify the types of cartilage defects.

Line 93: please add a specific reference.

Line 96: Please explain the abbreviation BMI.

Line 102: please explain how this cutoff has been chosen.

Line 103: this reference is not specific. Please replace with a more specific one.

Line 106: Please add the complete information (name, city, country, etc.) about the MRI used.

Line 115: References is lacking. Please add.

2.1. Surgical Procedure

Line 136: Please add the complete information (name, city, country, etc.) about the materials used.

Figure 2 is not clear. Please modify or substitute this figure.

2.3 Statistical analysis

Line 158: SPSS software has to be cited correctly, as it is reported in the web site.

Lines 159-160: please clarify this sentence.

Lines 164-165: please clarify this sentence.

Results

This section has to be improved better reporting the results of the whole cohort and of the two groups. It is difficult to read the manuscript without a comprehensive visualization of the results in specific and organised tables/figures.

It should be useful to report the results of the clinical outcomes at baseline (pre-operatively) and at different follow-up in a table along with p-values.

Decimal comma should be not used; please correct.

A flowchart, according to the STROBE criteria, could be useful to better understand patients’ recruitment.

Table 1: this table has to be modified according to the MDPI instructions.

This table should include the demographic and clinic information of the whole cohort and of the two different groups. The number of patients of each group has to be reported in the table. Please report that group A includes patients aged < 45 years and group B > 45 years. Mean age has to reported with standard deviation. Mean area was reported as mm2 in the table but in cm2 in the text. Please correct.

Figure 3: abbreviations have to explained in the caption of the figure. Please add which data are reported in axis x and y. Please add figures titles. Please add significant p-values.

Lines 215-225: R values should be reported for all correlations along with p-values. Only statistically significant correlations should be reported.

Discussion

This section has to be improved better discussing the results have to be discussed in accordance or not with the literature.

Lines 290-291. Please add the small number of patients in the limit of the study.

Comments on the Quality of English Language

English has to be revised       

Reviewer 3 Report

Comments and Suggestions for Authors

Round 2

Reviewer 2 Report

Comments and Suggestions for Authors

Abstract

Lines 42: when authors report a correlation, R value and its p-value of correlation have to be reported. Please correct.

Introduction

Lines 62-63 are a repetition of lines 60-62. Please correct.

Materials and methods

Line 89: please specify the types of cartilage defects.

Line 129: a specific reference about the Outerbridge classification is still lacking. Please add.

Line 103: references for Lysholm score and VAS are not specific. Please replace with more specific ones about the two scores.

Line 152 – reference 18 has to be reported also at this line.

2.3 Statistical analysis

Line 158: SPSS software has to be cited correctly, as it is reported in the web site.

Line 222: authors have to report how normality was checked.

Line 224: association should be replaced by correlation (correlation and association are two different concepts).

Lines 227-228: this sentence has to be better clarified.

Results

Decimal comma should be not used when reporting numbers in English language; please substitute with dots. E.g. line 277: substitute 3,0±1,2 with 3.0±1.2

Table 1: The column “mean age (y.o.)” and the column “mean area of defect” have to report also the standard deviation. Please correct.

Lines 306-308: when authors refer to correlations, R values have to be reported along with their specific p-values.

Figure 4

Part (b) shows MOCART score in both groups at one year and two years F-Up and not “show satisfactory clinical and instrumental results in both groups”, as reported by the authors. Please correct.

Significant p-values have to be added.

Please add which data are reported in axis y.

Lines 266-271: authors reported different values compared to the previous version. Please clarify.

Table 2 has to be modified in order to be clear; in this form is not understandable which comparisons have been done. In other words, it is not clear which comparisons are referred to the p-values.

Moreover, did authors compare the two different groups at different time points? Please clarifiy.

Discussion

This section usually begins with a brief introduction of the topic. Then, the aims of the study have to be reported and not the conclusions.  Please modify.

Comments on the Quality of English Language

Minor editing is required.
